# Lipopeptide Epimers and a Phthalide Glycerol Ether with AChE Inhibitory Activities from the Marine-Derived Fungus *Cochliobolus Lunatus* SCSIO41401

**DOI:** 10.3390/md18110547

**Published:** 2020-10-30

**Authors:** Yu Dai, Kunlong Li, Jianglian She, Yanbo Zeng, Hao Wang, Shengrong Liao, Xiuping Lin, Bin Yang, Junfeng Wang, Huaming Tao, Haofu Dai, Xuefeng Zhou, Yonghong Liu

**Affiliations:** 1CAS Key Laboratory of Tropical Marine Bio-resources and Ecology, Guangdong Key Laboratory of Marine Materia Medica, South China Sea Institute of Oceanology, Chinese Academy of Sciences, Guangzhou 510301, China; daiyu15@mails.ucas.ac.cn (Y.D.); likunlong16@mails.ucas.edu.cn (K.L.); shejianglian20@mails.ucas.ac.cn (J.S.); ljrss@126.com (S.L.); xiupinglin@hotmail.com (X.L.); yangbin@scsio.ac.cn (B.Y.); wangjunfeng@scsio.ac.cn (J.W.); 2Southern Marine Science and Engineering Guangdong Laboratory (Guangzhou), Guangzhou 511458, China; 3Research and Development of Natural Product from Li Folk Medicine, Institute of Tropical Bioscience and Biotechnology, Chinese Academy of Tropical Agriculture Sciences, Haikou 571101, China; zengyanbo@itbb.org.cn (Y.Z.); wanghao@itbb.org.cn (H.W.); daihaofu@itbb.org.cn (H.D.); 4School of Traditional Chinese Medicine, Southern Medical University, Guangzhou 510515, China; taohm@smu.edu.cn; 5Wuya College of Innovation, Shenyang Pharmaceutical University, Shenyang 110016, China

**Keywords:** lipopeptides, phthalide, *Cochliobolus lunatus*, absolute configurations, AChE inhibitory

## Abstract

A pair of novel lipopeptide epimers, sinulariapeptides A (**1**) and B (**2**), and a new phthalide glycerol ether (**3**) were isolated from the marine algal-associated fungus *Cochliobolus lunatus* SCSIO41401, together with three known chromanone derivates (**4**–**6**). The structures of the new compounds, including the absolute configurations, were determined by comprehensive spectroscopic methods, experimental and calculated electronic circular dichroism (ECD), and Mo_2_ (OAc)_4_-induced ECD methods. The new compounds **1**–**3** showed moderate inhibitory activity against acetylcholinesterase (AChE), with IC_50_ values of 1.3–2.5 μM, and an in silico molecular docking study was also performed.

## 1. Introduction

In order to survive in a highly competitive environment, many marine organisms can produce an array of biologically-active secondary metabolites, many of which exert cytotoxic or chemical defense activity [1]. The production of metabolites targeting cholinesterase enzymes among sessile organisms—such as sponge, coral, and algae—represent an additional advantage in a survival strategy against different predators and/or fouling invertebrates [2].

Cholinesterase enzymes, such as acetylcholinesterase (AChE), are widely expressed throughout the animal kingdom, from land to sea. AChE is the key enzyme for the termination of neurotransmission in cholinergic pathways, and AChE inhibition is an effective approach for the symptomatic treatment for Alzheimer’s disease (AD) [3,4].

In our recent efforts to search for bioactive natural products from marine-derived fungi, one endophytic fungus—*Cochliobolus lunatus* SCSIO41401—was isolated from the alga *Coelarthrum* sp., collected in Yongxing Island, in the South China Sea. In our previous work, several cytotoxic and antibacterial eremophilane sesquiterpenes [5] and anti-influenza spirostaphylotrichins [6] were discovered from this fungus SCSIO41401, when it was fermented with liquid medium. In order to discover more diverse and bioactive compounds from this strain, a fermentation with a solid rice medium was recently undertaken, and six natural products were obtained from this strain for the first time (Figure 1). Three of them were identified as new compounds with AChE-inhibitory activity. Described herein are the isolation, structure determination, and biological evaluation of these compounds.

## 2. Results

### 2.1. Structure Elucidation

Compound **1** was obtained as a brown oil. The HRESIMS (*m*/*z* 418.1951 [M + Na]^+^) data suggested a molecular formula of C_19_H_29_N_3_O_6_, revealing seven degrees of unsaturation. The analysis of the NMR data showed the presence of three methyls (including one oxygenated), seven methylenes (including one oxygenated), four methines (including one olefinic), one olefinic carbon, and four carbonyl carbons (Table 1). Combined with the 1D-NMR data, the ^1^H-^1^H COSY correlations of H-9/H-10/H-11/H-12 and the HMBC correlations from H-9 to C-12/C-8, H-11 to C-13, and H-12 to C-8/C-13 indicated the presence of one Pro residue. On the other hand, the ^1^H-^1^H COSY correlations of H-14/H-15/H-16/H-17, and the HMBC correlations of H-14 with C-13/C-17, H-16 with C-19, H-17 with C-13/C-19, and H-18 to C-14 suggested the presence of another amino acid residue, 5-methoxy-prolinamide (Me-O-Pro-NH_2_). Moreover, the HMBC correlations of H-11/H-12/H-14/H-17 with C-13 confirmed the connection of two amino acid residues. The ^1^H-^1^H COSY correlations from H-3 to H-4, and the HMBC correlations from H-1/H-3 to C-2, H-4/H-6 to C-5, H-7 to C-4/C-5/C-6 displayed the presence of the branched chain. The HMBC correlations from H-6/ H-9/H-12 to C8 showed the connection of the branched chain with the C-8 amino acid group. Thus, the planar structure of **1** was determined, as shown in Figure 2.

Compound **2** was isolated as a brown oil. Its molecular formula was reported to be same as **1** by HRESIMS (*m*/*z* 418.1958 [M + Na]^+^), which required seven degrees of unsaturation. The ^1^H-^1^H COSY correlations of H-9/H-10/H-11/H-12, and the HMBC correlations of H-9 with C-12/C-8, H-11 with C-13, and H-12 with C-8/C-13 indicated the presence of one Pro residue. In addition, the ^1^H-^1^H COSY correlations of H-14/H-15/H-16/H-17, and the HMBC correlations of H-14 with C-13/C-17, H-16 with C-19, H-17 with C-13/C-19, and H-18 to C-14 suggested the presence of another amino acid residue, 5-methoxy-prolinamide (Me-O-Pro-NH_2_). All of those indicated that compounds **1** and **2** were a pair of epimers.

The configurations of the Δ^5^ double bond of **1** and **2** were deduced as *E*, based on the ROESY correlation of H_2_-4/H-6 (Figure 2). The ROESY correlation of H-14 with H-17 indicated that H-14/H-17 were co-facial in **1**. In order to further determine the absolute configuration of C-12/C-14/C-1**7** of **1,** ECD calculations with four truncated models—12*R**,14*S**,17*S****-1** and 12*S**,14*S**,17*S**-**1**—were used. Based on the time-dependent density functional theory (TD-DFT), the Boltzmann-weighted ECD curves of 12*R*,14*S*,17*S***-1** gave the best agreement (Figure 3A), which led to the determination of the 12*R*,14*S*,17*S* absolute configurations of **1.** Comparing the NMR spectra of **1** and **2**, the ROESY correlation of H-14 with H-17 was not observed in **2**. Additionally, the difference in the coupling constant and the splitting of the peaks of H-14 indicated that **2** should be an epimer of **1** at C-14. From a biosynthetic point of view, the absolute configurations of **2** were suggested to be 12*R**,14*R**,17*S**. The Boltzmann-weighted ECD curves of 12*R*,14*R*,17*S*-**2** gave the best agreement with the experimental CD (Figure 3B). Thus, the structures of the two lipopeptide epimers were identified as shown in Figure 1, and were named sinulariapeptides A (**1**) and B (**2**).

Compound **3** was obtained as a brown oil. The molecular formula was determined as C_13_H_16_O_6_ by HRESIMS (*m*/*z* 291.0838 [M + Na]^+^). The NMR spectra indicated the presence of one methy, three methenes, four methines (including two olefinic and two oxygenated), four aromatic quaternary carbons, and a ketone carbonyl carbon (Table 1). A comparison of its ^1^H and ^13^C NMR data with those of (*S*)-3-ethyl-6,7-dihydroxyphthalide, another phthalide compound we obtained in this strain previously [5], indicated that they shared the same substructure [7,8]. The difference was the replacement of a glycerol ether substituent at C-7, which was further confirmed by the COSY correlations from H-4/H-5; H-12/H-13/H-14 (Figure 2), and the HMBC correlations (Figure 2) from H-12 to C-7. Thus, the planar structure of **3** was established.

The absolute configurations of **3** were determined by the CD method. The CD of the α,β-unsaturated γ-lactone rings with a chiral γ-carbon shows Cotton effects associated with the π→π* transition in the region 200–235 nm, and the n→π* transition in the region 235–270 nm [9,10,11,12]. Compound **3** showed a positive π→π* Cotton effect at 217 nm, and a negative n→π* Cotton effect at 259 nm, indicating the *S* absolute configuration at C-3 (Figure 3C) [9,10]. The stereogenic center of C-13 was characterized by the dimolybdenum induced CD (ICD) analysis. In the ICDs conducted by Snatzke’s method [13,14] using dimolybdenum tetraacetate (Mo_2_(OAc)_4_) in DMSO, the Mo_2_-complex of **3** gave positive CD bands II (408 nm) and IV (around 313 nm) (Figure 3D), and confirmed that C-13 was (*R*)-configured. Finally, the absolute configurations of **3** were assigned as 3*S*, 13*R*.

In addition, the structures of three known compounds (**4**–**6**) were elucidated by the analysis of the spectral data (Appendix A), as well as through comparison with those reported in the literature. They were identified as 2-(2’-Hydroxypropyl)-5-methyl-7-hydroxychromone (**4**) [15], orthosporin (**5**) [16], and bipolarinone (**6**) [17], respectively (Figure 1). All of these compounds are benzopyrone derivates, while compound **6** contains two benzopyrone moieties.

### 2.2. Bioassays

The new compounds **1**–**3** were evaluated for their cytotoxicities against several human cancer cell lines: K562, BEL-7402, SGC-7901, A549, and Hela. However, none of the compounds showed obvious inhibitory activity against those cells at a concentration of 50 μM in the preliminary screening (an inhibition rate less than 30%).

In the screening assay of the AChE inhibitory activities, compounds **1**–**3** showed moderate inhibitory activity, with IC_50_ values of 1.8 ± 0.12, 1.3 ± 0.11, and 2.5 ± 0.21 μM, respectively, in which huperzine A was used as a positive control (IC_50_ 0.30 ± 0.06 μM). The preliminary kinetic study of **1** towards AChE suggested the noncompetitive inhibition mode (Appendix A).

### 2.3. Molecular Docking

In order to gain an insight into the molecular interactions between compounds **1**–**3** and AChE, the crystal structure of the recombinant human Acetylcholinesterase enzyme (rhAChE, PDB ID: 4EY7) [18] was selected and subjected to an in silico molecular docking analysis with **1**–**3**, using the induced-fit module in the Schrödinger software suite [19]. As the docking results showed in Figure 4A, **1**–**3** were obviously able to bind to the active pocket of AChE, with TYR124 as the center of the binding site of AChE. The carbonyl group at C-8 connected to the Pro residue in the structures of **1** and **2** formed a hydrogen bond interaction with the active site residue TYR124 of AChE. Although the CONH_2_-19 group of **1** and **2** formed a hydrogen bond interaction with the different residues, SER125 and GLU202, respectively, there is no significant difference in the docking effects of AChE with **1** and **2**. In the docking model of AChE and **3**, a hydrogen bond interaction was formed with the active site residues TYR124 and PHE295. Moreover, the phthalide ring of **3** played a key role to form a π−π stacking interaction with TRP286.

## 3. Materials and Methods

### 3.1. General Experimental Procedures

The UV spectra were recorded on a UV-2600 UV-Vis spectrophotometer (Shimadzu, Japan). The optical rotations were measured using a PerkinElmer MCP-500 Polarimeter (Anton, Austria). The HRESIMS data were recorded on a Bruker maXis Q-TOF in positive/negative ion mode (Bruker, Fällanden, Switzerland). The NMR spectra were obtained on a Bruker Avance spectrometer (Bruker) operating at 500 and 700 MHz for ^1^H NMR, and 125 and 175 MHz for ^13^C NMR, using tetramethylsilane (TMS) as an internal standard. The chemical shifts were given as *δ* values, with J values reported in Hz. TLC plates with silica gel GF254 (0.4–0.5 mm, Qingdao Marine Chemical Factory, Qingdao, China) were used for the analytical and preparative TLC. The column chromatography was carried out on silica gel (200–300 mesh, Jiangyou Silica Gel Development Co., Yantai, China), YMC Gel ODS-A (12 nm, S-50 μm YMC, MA, USA) and Sephadex LH-20 (40–70 μm, Amersham Pharmacia Biotech AB, Uppsala, Sweden). The semi-preparative HPLC was carried on HTACHI L2130 with YMC ODS SERIES (YMC-Pack ODS-A, 250 × 10 mm I.D., S-5 μm, 12 nm) and the analysis HPLC was carried on a shimadzu LC-10ATvp with YMC ODS SERIES (YMC-Pack ODS-A, 250 × 4.6 mm I.D., S-5 μm, 12 nm). Spots were detected on the TLC under UV light, or by heating after spraying with the mixed solvent of saturated vanillin and 5% H_2_SO_4_ in H_2_O. The artificial sea salt was a commercial product (Guangzhou Haili Aquarium Technology Company, Guangzhou, China).

### 3.2. Fungal Material

The fungal strain SCSIO41401 was isolated from a marine alga *Coelarthrum* sp. collected in Yongxing Island, South China Sea [5]. The strain was identified as *Cochliobolus lunatus* SCSIO41401, according to the ITS region sequence. The producing strain was stored on Medium B (malt extract: 15 g, sea salt: 24.4 g, agar: 15 g, water: 1 L and pH: 7.4–7.8) agar slants at 4 °C, and then deposited at the Key Laboratory of Tropical Marine Bio-resources and Ecology, Chinese Academy of Science.

### 3.3. Fermentation and Extraction

The seed culture was prepared by inoculating spores of the strain *Cochliobolus lunatus* SCSIO41401 into a 1000 mL flask containing 10 mL seed medium (malt extract: 15 g, sea salt: 10 g, distilled water: 1 L and pH: 7.4–7.8), and incubated at 25 °C on a rotary shaker (178 rpm) for 3 days. In total, 10 mL seed culture was then transferred into 1 L × 48 conical flasks with solid rice medium (each flask contained 200 g rice, 2.5 g sea salt and 210 mL naturally-sourced water), and the large scale fermentation of the strain was carried out at 25 °C for 30 days. The total rice culture was crushed and extracted with EtOAc three times. The EtOAc extract was evaporated under reduced pressure to afford a crude extract. The extract was suspended in MeOH and then partitioned with equivoluminal petroleum ether to separate the oil. Finally, the MeOH solution was concentrated, yielding a black extractive (256.4 g).

### 3.4. Extraction and Purification

The crude extract was subjected to silica gel column chromatography eluting with CH_2_Cl_2_/MeOH (*v*/*v* 100:0–0:100) to give twelve fractions (Fr.1–Fr.12). Fr.6 was subjected to reversed phase medium-pressure liquid chromatography (MPLC) with MeOH/H_2_O (*v*/*v* 2:8 to 10:0) to offer thirteen sub-fractions. Fr.6-4 was further purified by semi-preparative reverse phase (SP-RP) HPLC with MeOH/H_2_O (*v*/*v* 45:55, 2.0 mL/min) to yield compound **5** (t_R_ = 27.0 min, 105.8 mg, with purity of 98.5%). Fr.6-11 was purified by SP-RP HPLC with MeOH/H_2_O (*v*/*v* 57:43, 3.0 mL/min) to yield compound **6** (t_R_ = 30.0 min, 6.2 mg, with a purity of 95.5%). Fr.7 was subjected to reversed phase MPLC with MeOH/H_2_O (*v*/*v* 1:9 to 10:0) to offer fifteen sub-fractions. Fr.7-8 was further purified by semi-preparative reverse phase (SP-RP) HPLC with CH_3_CN/H_2_O (*v*/*v* 25:75, 2.0 mL/min) to yield compounds **4** (t_R_ = 12.8 min, 10.4 mg, with a purity of 98.0%), **2** (t_R_ = 15.7 min, 7.0 mg, with a purity of 97.8%), **1** (t_R_ = 18.6 min, 13.1 mg, with purity of 98.4%), and **3** (t_R_ = 26.5 min, 6.3 mg, with purity of 98.0%).

Compound **1**: brown oil; $[\alpha {]}_{D}^{25}$ +5.1 (c 0.198, MeOH); UV (MeOH) λmax (log ε) 210 (3.18) nm. For ^1^H and ^13^C NMR spectroscopic data, see Table 1; (+)-HRESEMS *m/z* 418.1951[M + Na] ^+^ (Calcd for C_19_H_29_NaO_6_, 418.1949) (Appendix A).

Compound **2**: brown oil; $[\alpha {]}_{D}^{25}$ −0.48 (c 0.101, MeOH); UV (MeOH) λmax (log ε) 200 (3.24) nm. For ^1^H and ^13^C NMR spectroscopic data, see Table 1; (+)-HRESEMS *m/z* 418.1958 [M + Na] ^+^ (Calcd for C_19_H_29_NaO_6_, 418.1949) (Appendix A).

Compound **3**: brown oil; $[\alpha {]}_{D}^{25}$ +2.4 (c 0.062, MeOH); UV (MeOH) λmax (log ε) 204 (3.10), 310 (2.26) nm; CD (0.33 mg/mL, MeOH) λmax (Δε) 217 (0.79), 259 (−0.07), 291 (0.15) nm. For ^1^H and ^13^C NMR spectroscopic data, see Table 1; (+)-HRESEMS m/z 291.0838 [M + Na] ^+^ (Calcd for C_13_H_16_NaO_6_, 291.0839) (Appendix A).

### 3.5. ECD Calculations for the Truncated Models of 1 and 2

The relative configurations of **1** and **2** were established initially on the basis of their ROESY spectra. The Molecular Merck force field (MMFF) and density functional theory (DFT)/time-dependent density functional theory (TDDFT) calculations of **1** and **2** were performed with the Spartan’14 software and Gaussian 09 software, respectively, using default grids and convergence criteria. A MMFF conformational search generated low energy conformers with a Boltzmann population of over 5% (the relative energy within 6 kcal/mol) [20], which were subjected to geometry optimization using the DFT method at the B3LYP/6-31G(d,p) level in MeOH using the conductor-like polarizable continuum model (CPCM). The overall theoretical calculation of the ECD was conducted in MeOH using time-dependent density functional theory at the B3LYP/6-31 + G(d,p) level for the stable conformers of **1** and **2**. The rotatory strengths for a total of 30 excited states were calculated. The ECD spectra of the different conformers were generated using the programmes SpecDis 1.6 (University of Würzburg, Würzburg, Germany) and Prism 5.0 (GraphPad Software Inc., La Jolla, CA, US) with a half-bandwidth of 0.3−0.4 eV, according to the Boltzmann-calculated contribution of each conformer after the UV correction.

### 3.6. Bioactivity Assay

Five human cancer cell lines—K562, BEL-7402, SGC-7901, A549, and Hela—were purchased from Shanghai Cell Bank, Chinese Academy of Sciences. Compounds **1**–**3** were evaluated for their cytotoxicities against those cell lines according to the reported methyl thiazolyl tetrazolium (MTT) method [5]. Briefly, the cancer cells were cultured in RPMI1640 media supplemented with 10% phosphate-buffered saline (FBS). The cells were seeded at a density of 400 to 800 cells/well in 384-well plates, and then incubated with the tested compounds in one concentration of 50 μM for the preliminary screening, with 5-flurouracil as the positive control. After 72 h treatment, MTT reagent was added, and the OD value of each well was measured at 570 nm with an Envision 2104 multilabel reader (PerkinElmer). If the tested compound showed an obvious inhibitory activity with an inhibition rate of more than 30% in the preliminary screening test, the further screening with more concentrations was then performed. However, none of the compounds showed significant activity in the preliminary screening.

The inhibitory effects of compounds **1**–**3** on acetylcholinesterase (AChE, human-recombinant, purchased by Sigma-Aldrich, St. Louis, MO, USA, Catalog Number C1682) were measured in vitro in 96-well plates according to the modified Ellman methods [21]. Briefly, 0.2 Units of AChE were dissolved in 0.1M potassium phosphate buffer (pH 7.4), and compounds **1**–**3** dissolved in DMSO (with final concentrations of 8, 4, 2, 1, 0.5, 0.25 μM) were added to each well of a 96-well plate. Acetylthiocholine iodide and 5,5′-dithiobis (2-nitrobenzoic acid) were then added for a final concentration of 50 μM. The reaction was carried out for 30 min at 30 °C. The absorbance was measured at 410 nm by an Envision 2104 multilabel reader (PerkinElmer), and the IC_50_ value was calculated using Prism 5.0 (GraphPad Software Inc.). Huperzine A was used as the positive control, with an IC_50_ value of 0.30 ± 0.06 μM.

### 3.7. Molecular Docking Analysis

The Schrödinger 2017-1 suite (Schrödinger Inc., New York, NY, USA) was employed to perform the docking analysis [19]. The structure of AChE (PDB code: 4EY7) [18] was used as a starting model—with all of the waters and the N-linked glycosylated saccharides removed—and was constructed following the Protein Prepare Wizard workflow in Maestro 11-1. The prepared ligands were then flexibly docked into the receptor using the induced-fit module with the default parameters. The figures were generated using PyMol molecular graphics software (Schrödinger 2017-1, Schrödinger Inc., New York, NY, USA).

## 4. Conclusions

Three new compounds—a pair of lipopeptide epimers (**1** and **2**) and a phthalide glycerol ether (**3**)—were isolated from the marine algal-associated fungus *Cochliobolus lunatus* SCSIO41401. Their structures, including their absolute configurations, were determined by comprehensive spectroscopic methods, together with experimental, calculated, and Mo_2_ (OAc)_4_-induced ECD methods. The in vitro bioassay and in silico docking study revealed compounds **1**–**3** to be moderate AChE inhibitors.

## Figures and Tables

**Figure 1 marinedrugs-18-00547-f001:**
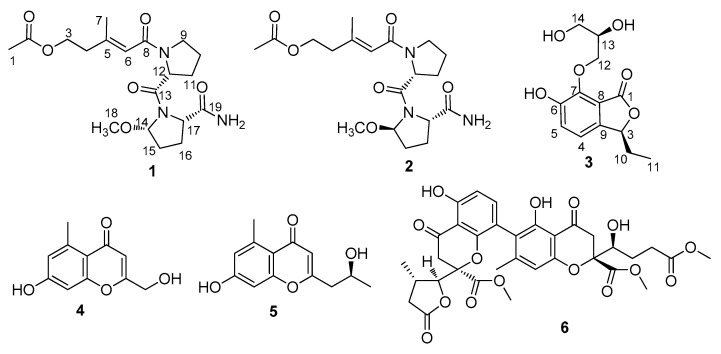
Compounds **1**–**6**, isolated from fungal strain SCSIO41401.

**Figure 2 marinedrugs-18-00547-f002:**
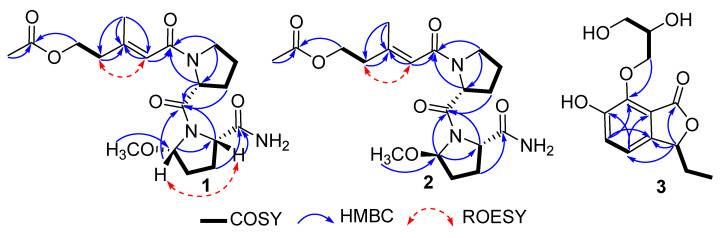
The key HMBC, COSY, and ROESY correlations of **1**–**3**.

**Figure 3 marinedrugs-18-00547-f003:**
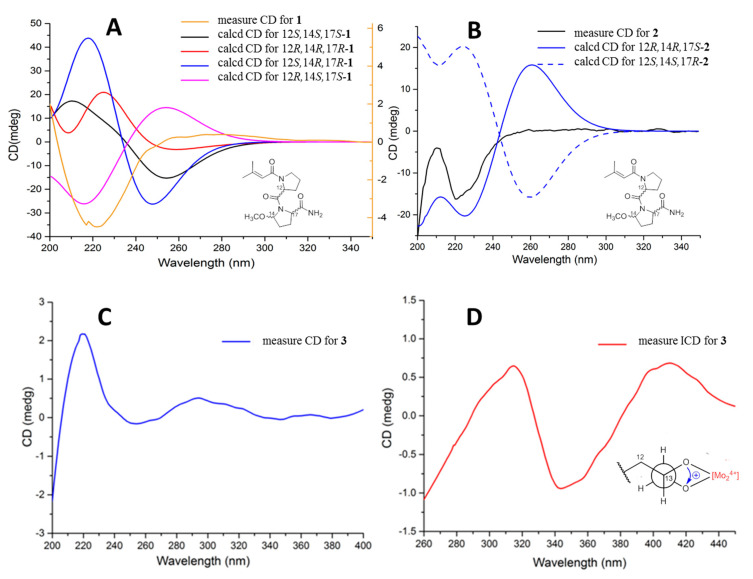
(**A**) Experimental CD spectra of **1,** and the calculated ECD spectra of its truncated models. (**B**) Experimental CD spectra of **2**, and the calculated ECD spectra of its truncated models. (**C**) Experimental CD spectra of **3** in MeOH. (**D**) Induced CD (ICD) spectra from the Mo_2_-complexes of **3** in DMSO.

**Figure 4 marinedrugs-18-00547-f004:**
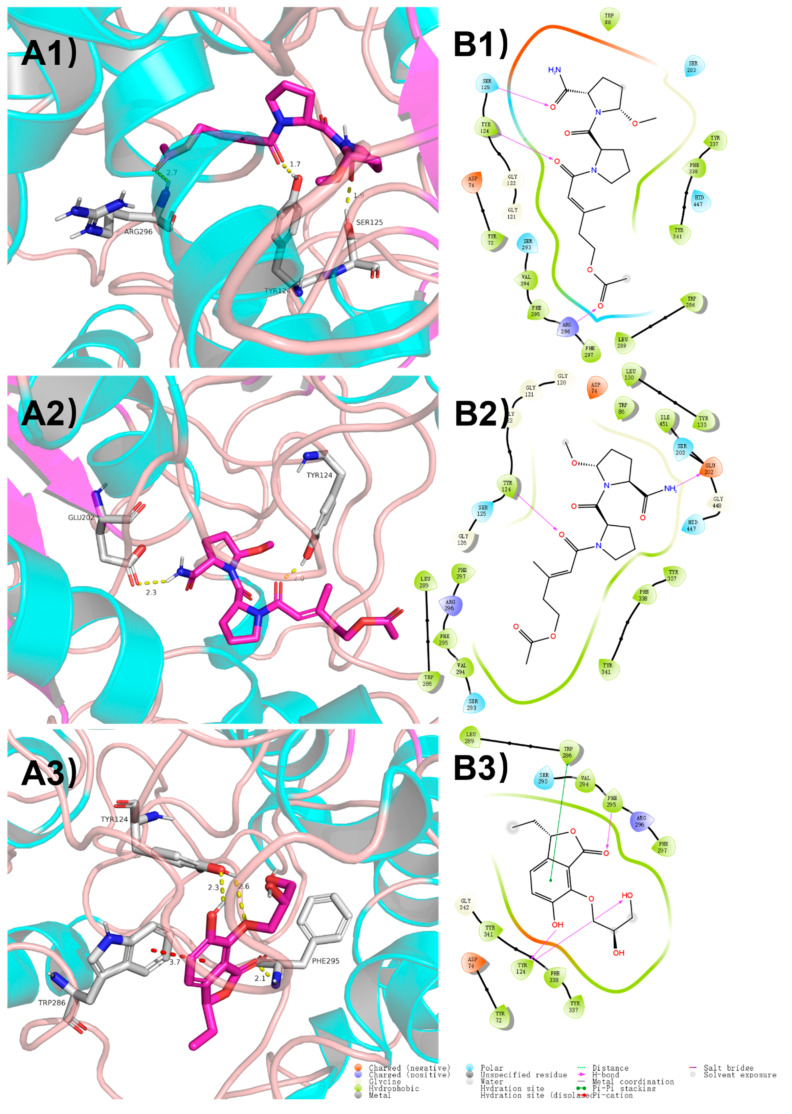
Molecular docking of **1**–**3** with AChE (PDB code: 4EY7). The binding sites of the molecules **1** (**A1**), **2** (**A2**) and **3** (**A3**) with the AChE protein. The 2D interaction details of the predicted binding mode of **1** (**B1**), **2** (**B2**) and **3** (**B3**) with the AChE.

**Table 1 marinedrugs-18-00547-t001:** ^13^C and ^1^H NMR data for **1**–**3** in CD_3_OD (*δ*_C_ and *δ*_H_ are given in ppm).

Pos.	1		2		3	
	*δ* _C_ ^a^	*δ*_H_, (*J* in Hz) ^b^	*δ* _C_ ^c^	*δ*_H_, (*J* in Hz) ^d^	*δ* _C_ ^c^	*δ*_H_, (*J* in Hz) ^d^
1	20.8 CH_3_	2.02 s	20.8 CH_3_	2.02 s	172.0 C	
2	173.0 C		172.9 C			
3	63.3 CH_2_	4.22 t (6.5)	63.2 CH_2_	4.22 t (6.5)	83.3 CH	5.39 dd (7.0, 4.2)
4	40.4 CH_2_	2.43 t (6.5)	40.3 CH_2_	2.43 t (6.5)	118.5 CH	7.05 d (8.4)
5	150.7 C		150.7 C		127.1 CH	7.19 d (8.4)
6	121.2 CH	5.72 brd (1.0)	121.2 CH	5.73 brd (1.0)	156.1 C	
7	18.4 CH_3_	2.13 d (1.5)	18.4 CH_3_	2.13 d (1.5)	146.9 C	
8	169.5 C		169.5 C		140.8 C	
9	39.8 CH_2_	3.23 t (7.0)	39.8 CH_2_	3.24 t (6.3)	119.3 C	
10	26.0 CH_2_	1.63 m	25.4 CH_2_	1.66 m	28.9 CH_2_	2.11 m, 1.79 m
11	27.7 CH_2_	1.93 m/1.81 m	27.2 CH_2_	1.98 m/1.84 m	8.9 CH_3_	0.97 t (7.0)
12	55.8 CH	4.24 t (5.0)	56.4 CH	4.14 t (4.2)	77.0 CH_2_	4.31 dd (10.5, 3.5) 4.19 dd (10.5, 6.3)
13	169.3 C		170.4 C		72.0 CH	3.95 m
14	89.3 CH	5.44 t (8.0)	88.5 CH	5.26 d (4.9)	63.9 CH_2_	3.75 m, 3.69 m
15	31.0 CH_2_	1.87 m	31.7 CH_2_	1.90 m		
16	26.1 CH_2_	2.34 m	25.9 CH_2_	2.16 m 2.11 m		
17	58.7 CH	4.37 t (7.0)	60.5 CH	4.24 t (7)		
18	56.8 CH_3_	3.38 s	57.4 CH_3_	3.37 s		
19	172.4C		173.5 C			

^a^ in 125 MHz, ^b^ in 500 MHz, ^c^ in 175 MHz, ^d^ in 700 MHz.

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
