# Peer review of "Lipopeptide Epimers and a Phthalide Glycerol Ether with AChE Inhibitory Activities from the Marine-Derived Fungus Cochliobolus Lunatus SCSIO41401"

_marinedrugs, 2020, doi:10.3390/md18110547_

Round 1

Reviewer 1 Report

Dai et al obtained two lipopeptide epimers from endophytic fungus SCSIO41401. They determined the structures of these compounds and their moderate inhibitory activity against acetylcholinesterase. The following details are important and were  not clear described in the draft:

  1. The initial purity of compounds (i.e. brown oil)  1 and 3 are not shown. The oils may contain several different materials in equal amounts.
  2. Compound 2 did not described. And its characterization exactly like for compound 1 (see P.1) is required.
  3. Bioassays described very poor. The details of experiments, such as how the treatments was provided, the time of incubation of the cells in the presence of  epimers  etc have to be described in details.
  4. The inhibition curves for all checked cells have to be shown is SI.
  5. It does not clear how the authors provided the inhibition of AChE . The experiments has to be described in details.
  6. All the kinetic curves have to be shown. The method for IC50 calculation has to be described.

Author Response

According to the reviewers’ comments, we have revised the relevant part in the original manuscript, and respond the comments point-by-point, listed below. The modifications in the manuscript are highlighted in the text.

  1. Thanks for your kind suggestion. We have indicated the purity of the compound in the part 3.4. The purity of all those compounds was higher than 95%, analyzed by HPLC. The 1H-NMR spectra also showed their eligible purity. (in Lines 191-197)
  2. We have added the structural characterization of 2 in the revised manuscript. (in Lines 73-80)

  3. The bioassay in the original manuscript is very brief, as consideration is the usual method. The Reviewer 1' opinions are very good. We have added the method and details of activity test in the revised manuscript. (in Lines 127-129, 224-242)

  4. In our study, compound with only one concentration of 50 μM was used for preliminary screening. If the tested compound shows the obvious inhibitory activity with the inhibition rate more than 30% in the preliminary screening test, the further screening with more concentrations will be taken. However, none of the compounds showed significant activity in the preliminary screening. So the inhibition curves could not been shown here.

  5. We have added the method and details of AChE activity test in the revised manuscript. (in Lines 233-242)
  6. The IC50 value was calculated using Prism 5.0 (GraphPad Software Inc.). (in Line 241) We plan to perform the mode of AChE inhibitory action study, as well as kinetic curves, of compounds 1-3, but the amount of compounds was no longer sufficient. Although it is a pity that we cannot carry out further experiments to enrich our study, this result of our AChE inhibitory test in this paper should also be valuable.

Reviewer 2 Report

The article by Dai et al. claims to provide chemical information pertaining to six compounds isolated from a fungal strain.  Additionally, the authors claim to provide evidence of these compounds inhibiting acetylcholinesterase in the low micromolar range.  Acetylcholinesterase inhibitors have been seen as therapeutic agents for age related diseases in the last few years.  Hence, this is an active area of research interest among the general scientific community.  The overall isolation and characterization of the compounds seems adequate.  However, there are some major issues that should be addressed related to the compounds inhibition of acetylcholinesterase to make this paper stronger.

  1. Species of acetylcholinesterase tested. The authors tested the compounds with AChE from the electric eel instead of human AChE. AChE from different species has been observed to behave differently in terms of inhibition and reactivation.  As a result, inhibitory data related to the electric eel lack appreciably significant. Additionally, the authors model the compounds with human AChE.  The authors should really test their compounds against human AChE that is widely available in quantifies for enzymatic testing.
  2. Promiscuous inhibition. The authors provide no data related to selective inhibition of AChE.  These six compounds could easily be non-selective inhibitors. Additional experiments would need to be performed. For instance, a mode of action study.  Based on their model prediction, the compounds should behave in a competitive manner.  
  3. Figure 4. The there is no appreciable value to Figure 4 at the current resolution and point of view the images are rendered. Close up views of the inhibitor bound to the enzyme are necessary to gauge the value of the model and nature of the interactions.

Author Response

According to the reviewers’ comments, we have revised the relevant part in the original manuscript, and respond the comments point-by-point, listed below. The modifications in the manuscript are highlighted in the text.

  1. Thank you for your professional advice. In fact, the AChE enzyme we tested was from human (recombinant). I am very sorry that we miswrote another source of electric eels in the original manuscript. The human derived AChE we used was purchased by Sigma-Aldrich (Catalog Number C1682) (in Lines 233-234)
  2. Thanks for your kind suggestion. We couldn't agree with you more. A mode of inhibitory action study will make this paper stronger. At the beginning of our process of screening tests the obtained natural products, we only used the simplest and most convenient method, such as this AChE inhibitory test in this paper. We plan to perform the mode of inhibitory action study of compounds 1-3, but the amount of compounds was no longer sufficient. Although it is a pity that we cannot carry out further experiments to enrich our study, this result of our AChE inhibitory test in this paper should also be valuable, like many articles on Marine Drugs and J Nat Prod, reporting natural products with AChE inhibitory activities (in Lines 233-242)
  3. Thank you for your kind suggestions. We have amplificated in details of the 3D interaction in figure 4.

Reviewer 3 Report

This paper describes the isolation of two new lipopeptides and a phtalide glycerol ether with very interesting AChE inhibitory activities.

The structure elucidation by NMR experiments were assisted by the use of DFT TD-ECD methodology. Some minor mistakes are:

  1. Line 59:… seven methylenes.
  2. On figure 2, some of the key HMBC correlations mentioned in the text (discussion of the Structure elucidation of 1) are not included. Please add them: H9-C12, H14-C17, H4/H6 to C5, H7/C5.

A minor issue on lines 69 and 70, The HMBC correlations from H-9, H-12 to C8 showed the location of the fatty chain… is misleading.

The major problem that I found in the paper is the discussion of the absolute configuration of 1, 2 and 3.

Compounds 1 and 2: TD-ECD DFT calculations clearly showed the stereochemistry of 1 as 12R14R17S. However, for compound 2, just NMR differences between compunds 1 and 2 seem to explain a proposed 12R14R17S, epimer at C12 of 1. Surprisingly, no TD-CD curve for this compound was presented. Why so? Since all possibilities for the four pairs of diasteroisomers are represented on Figure 3, ECD of 2 should confirm the proposed stereochemistry. The CD curve for 2 is a must!

Compound 3: Also, I could'nt follow the assignment of the absolute stereochemistry for 3. The Mo complex would be formed between the hydroxyl groups at C-13 and C-14 (I am using the numbering used in Figure 1), so why the Newman projection in the complex in Figure 3C is drawn between C-12 (which contains two!! oxygens?) and some other carbon that I cannot able to identify… It makes no sense at all. What about the stereochemistry at C-3?  This part needs to be discussed properly.

Author Response

According to the reviewers’ comments, we have revised the relevant part in the original manuscript, and respond the comments point-by-point, listed below. The modifications in the manuscript are highlighted in the text.

Minor mistakes:

Thanks for your suggestion. We have changed the wrong words and missing HMBC correlations on the figure 2. (in Lines 146-149). For misleading sentences in Line 69 and 70, we have replace it with “The 1H-1H COSY correlations from H-3 to H-4, and the HMBC correlations from H-1/H-3 to C-2, H-4/H-6 to C-5, H-7 to C-4/C-5/C-6 displayed the presence of the branched chain. The HMBC correlations from H-6/ H-9/H-12 to C8 showed the connection of the branched chain with C-8 amino acid group.” (in Lines 69-71)

Major problem:

Thanks for your proposal. According to your kindly suggestion, the ECD curves of 12R*,14R*,17S*-2 were calculated and compared with the experimental ECD curve, which led to the determination of 12R,14R,17S absolute configurations of 2, an epimer of 1 at C-14. (in Lines 96-98 and Fig 3B)

And for compound 3, the Newman projection in the complex should be between C-13 (instead of C-12) and C-14. We are very sorry to miswrite the number in the original paper. (in Fig 3D) The stereochemistry of C-3 was determined by CD comparation. The CD of the α,β-unsaturated γ-lactone rings with a chiral γ-carbon shows Cotton effects associated with the π→π* transition in the region 200–235 nm and the n→π* transition in the region 235–270 nm, as reported in many references (references 9-12 in the manuscript). Compound 3 showed positive π→π* Cotton effect at 217 nm and negative n→π* Cotton effect at 259 nm, indicating the S absolute configuration at C-3. (in Lines 111-115 and Fig 3C)

Round 2

Reviewer 1 Report

The draft by Dai et al was significantly improved and I recommend to accept it.

Author Response

Thank you!

Reviewer 2 Report

The revised article by Dai et al. claims to provide chemical information pertaining to six compounds isolated from a fungal strain.  Additionally, the authors claim to provide evidence of these compounds inhibiting acetylcholinesterase in the low micromolar range.  Acetylcholinesterase inhibitors have been seen as therapeutic agents for age related diseases in the last few years.  Hence, this is an active area of research interest among the general scientific community.  Although the authors have effectively resolved this reviewer’s previous concerns about the sources of AChE tested and certain figures, they have not resolved a key concern.

Specifically, the authors do not address promiscuous inhibition. Without this data, the validity of the compounds is highly questionable. The authors response of compound availability does not bode well for reproducibility either.

Author Response

Thanks for your kind suggestion, together with your last comments. We indeed agree with you. A mode of inhibitory action study will make this paper stronger. As our response in last revised version, the amount of compounds is very limited to take the kinetic studies. Recently, we try to take the kinetic studies towards AChE with compound 1, which is the only compound barely feasible. The preliminary Lineweaver–Burk plots (Figure S33) for the inhibition of AChE with compound 1 were fitted to the noncompetitive inhibition mode in visual inspection.

Because the experiment is preliminary and the concentrations is not enough, the experiment has not been repeated, so we think this preliminary figure (Figure S33) is not suitable to appear in the manuscript. We present this preliminary figure in Supplementary and give a brief discussion in manuscript.

Reviewer 3 Report

Now the paper has significantly improved, and authors have addressed all the recomendations made. Therefore as it is, it merits to be published in Marine Drugs.

Author Response

Thank you!